# Progenitors oppositely polarize WNT activators and inhibitors to orchestrate tissue development

Irina Matos[1], Amma Asare[1†], John Levorse[1], Tamara Ouspenskaia[1‡], June de la Cruz-Racelis[1], Laura-Nadine Schuhmacher[2], Elaine Fuchs[1*]

[1]Robin Chemers Neustein Laboratory of Mammalian Cell Biology and Development, Howard Hughes Medical Institute, The Rockefeller University, New York, United States; [2]The Francis Crick Institute, Midland Road, London, United Kingdom

**Abstract** To spatially co-exist and differentially specify fates within developing tissues, morphogenetic cues must be correctly positioned and interpreted. Here, we investigate mouse hair follicle development to understand how morphogens operate within closely spaced, fate-diverging progenitors. Coupling transcriptomics with genetics, we show that emerging hair progenitors produce both WNTs and WNT inhibitors. Surprisingly, however, instead of generating a negative feedback loop, the signals oppositely polarize, establishing sharp boundaries and consequently a short-range morphogen gradient that we show is essential for three-dimensional pattern formation. By establishing a morphogen gradient at the cellular level, signals become constrained. The progenitor preserves its WNT signaling identity and maintains WNT signaling with underlying mesenchymal neighbors, while its overlying epithelial cells become WNT-restricted. The outcome guarantees emergence of adjacent distinct cell types to pattern the tissue.

*For correspondence:
fuchslb@rockefeller.edu

Present address: †Baylor College of Medicine, Houston, United States; ‡Department of Biology, Massachusetts Institute of Technology, Broad Institute of MIT and Harvard, Cambridge, United States

## Introduction

Embryonic development has long fascinated generations of scientists. Despite years of research, developmental biologists are still puzzled by the remarkable emergence of complex multicellular organisms from single cells. Central to understanding metazoan phenotypic reproducibility is the problem of pattern formation.

In the early 20th century, biologists began providing a new conceptual framework for understanding how cellular fates are specified during morphogenesis. Initially, it was proposed that depending upon their local concentration, 'materials' form gradients that dictate distinct patterning of otherwise uniform cellular sheets (*Boveri, 1901*; *Morgan, 1901*; *Dalcq, 1938*; *Rogers and Schier, 2011*). This notion began to crystallize in 1952, when Alan Turing applied mathematical modeling to explain how diffusion of two interacting chemical substances could spontaneously produce a pattern from an homogeneous field of cells (*Turing, 1952*; *Heller and Fuchs, 2015*).

Some years later, Lewis Wolpert posited the 'French Flag Problem' to describe a cell's differential gene expression according to its position within a morphogen gradient (*Wolpert, 1968*). He suggested that thresholds of morphogen gradients would establish boundaries that result in distinct cell fates. The 'positional information' model was then proposed to describe how complex patterns emerge from prior asymmetries (*Green and Sharpe, 2015*; *Wolpert, 1969*). The premise is that each cell has a positional value that specifies its position, and it is the interpretation of positional information that dictates cell fate (*Wolpert, 1989*).

Overall, these early studies popularized the view that morphogens and positional information function centrally in generating the symmetry-breaking events that differentiate cellular fates and drive morphogenesis. Despite these important advances on the establishment of two-dimensional

patterns, comparatively little is known about the molecular nature of the positional information needed to generate three-dimensional tissue patterns, or how closely juxtaposed cells within a developing tissue and organ adopt and maintain distinct cellular fates. Here, we tackle this problem by using the emergence of hair follicles in developing mammalian skin as a classical example of three-dimensional patterning in morphogenesis.

During embryonic development, the first of three spatially positioned arrays of hair placodes emerges when some cells within an epithelial monolayer begin to experience a higher level of WNT signal than their neighbors (*DasGupta and Fuchs, 1999*). Similar to *Drosophila* development, these WNTs act as short-range inducers and long-range organizers. Thus, through either rapid reaction-diffusion (*Sick et al., 2006*; *Glover et al., 2017*) or mechanotransduction-mediated mesenchymal self-organization (*Shyer et al., 2017*), the WNT$^{hi}$ cells within the plane of homogeneous epidermal cells cluster into an array of evenly spaced placodes (*Ahtiainen et al., 2014*). As placodes form, they produce inhibitory signals such as bone morphogenic proteins (BMPs) that limit placode size and distance placodes from each other (*Narhi et al., 2008*; *Noramly and Morgan, 1998*).

Three dimensional pattern formation begins when WNT signaling reaches a threshold in placode cells, stimulating them to divide perpendicularly relative to the epidermal plane and generating differentially fated progenitor daughters (*Ouspenskaia et al., 2016*). Intriguingly, these early basal daughters both produce WNTs and respond to WNTs, as exemplified by WNT-reporter activity and nuclear LEF1, a positive-acting downstream DNA binding effector of WNT-stabilized β-catenin (*Figure 1A*; *Ouspenskaia et al., 2016*). Interestingly, the overlying suprabasal daughter displays a paucity of WNT signaling and adopts a new fate, while the dermal condensate beneath the hair bud shows robust WNT signaling. How this positional information is locally and directionally partitioned and how sharp boundaries in WNT signaling are established between neighboring cells has remained elusive.

Here, we use mouse genetics to mosaically alter WNT signaling within basal progenitors of embryonic epidermis. By coupling transcriptome analyses with gain and loss of function studies, we unveil a cohort of WNT antagonists whose transcripts are WNT-sensitive and specifically activated in the WNT signaling basal progenitors. While morphogen inhibitors have been typically associated with negative feedback loops that either dampen or impair signaling, we find that even though they produce these inhibitors, basal progenitors still signal through WNTs. Moreover, they appear to do so by differentially polarizing activators and inhibitors to establish a spatially confined gradient within the placode. By perturbing it, we learn that this single-cell length morphogen gradient endows basal progenitors with the ability to orchestrate directional signaling. Progenitors generate a WNT-restricted microenvironment for their apical daughters, while fueling a basal basement membrane niche that is rich in WNT signaling at the epithelial-mesenchymal border.

## Results

### Sustained activation of WNT disrupts embryonic skin hexagonal patterning

In the skin, it is well-established that nuclear LEF1 co-localizes not only with nuclear β-catenin (*Fuchs et al., 2001*) but also with both *TOPGAL*, a WNT-reporter driven by an enhancer composed of multimerized LEF1 DNA binding sites (*DasGupta and Fuchs, 1999*), and as shown in *Figure 1A*, *Axin2-LacZ*, a WNT-reporter driven by the endogenous WNT target *Axin2* (*Lustig et al., 2002*). Thus, in this research, we often used nuclear LEF1 as a proxy for WNT signaling.

To begin to understand how WNT signaling promotes the symmetry-breaking events during skin development, we turned to our powerful in utero delivery method (*Beronja et al., 2010*). This method was superior over prior transgenic methods in that it allowed us to first, manipulate WNT signaling early, while the skin was still a single-layered epithelium, and second, generate mosaic perturbations in the signals that dictate hair follicle patterning (*Andl et al., 2002*).

To gain initial insights, we accentuated the WNT signaling response in skin patches by transducing a lentivirus (LV) harboring Cre recombinase into E9.5 mouse embryos floxed for *Adenomatous Polyposis Coli* (*Apc$^{fl/fl}$*). As expected from prior *Apc* loss of function studies on E14.5 embryos (*Kuraguchi et al., 2006*), mosaic loss of *Apc* resulted in overactivation of β-catenin/WNT signaling in patches of transduced skin (*Figure 1B*).

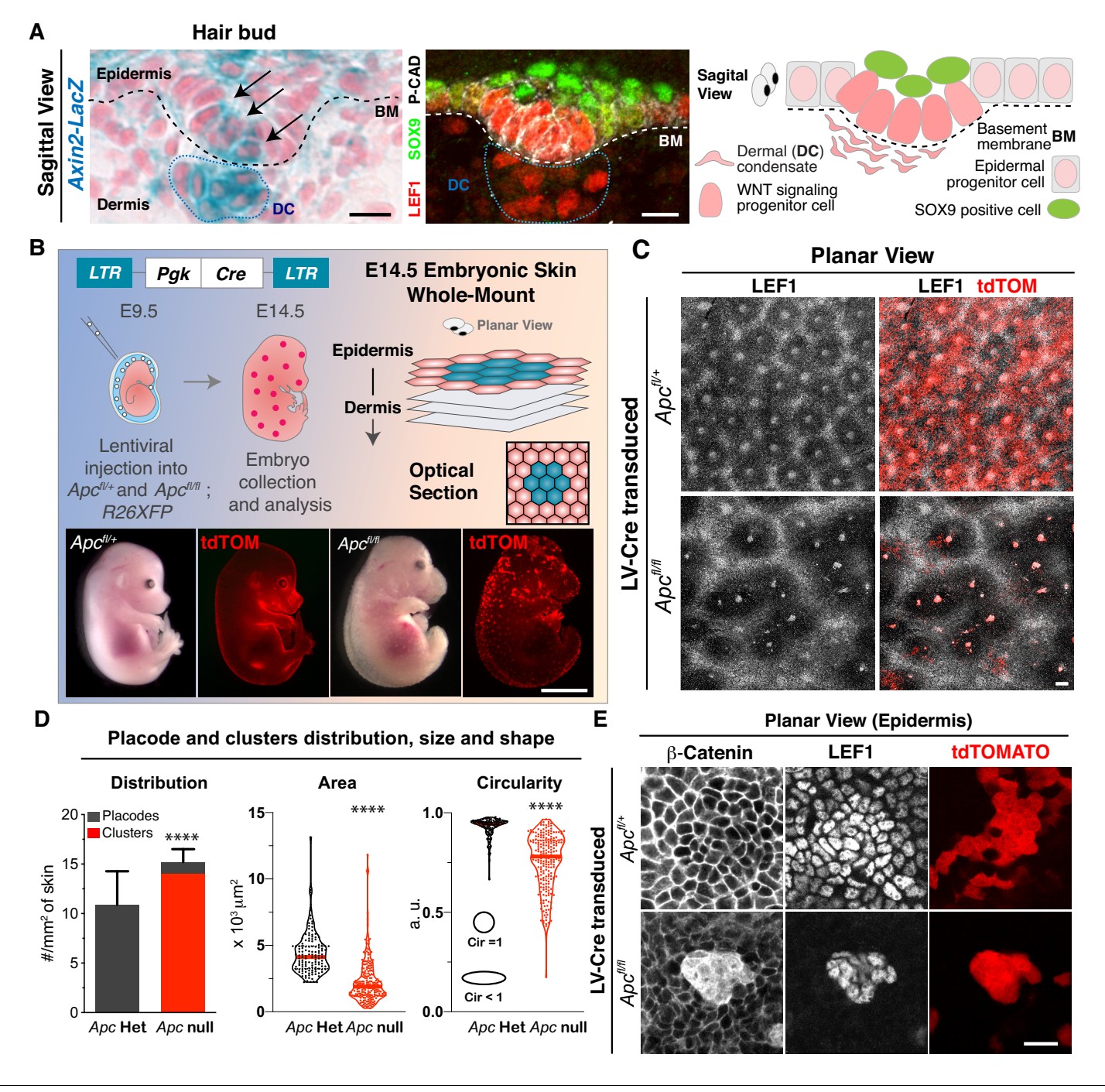

**Figure 1.** Two-dimensional patterning of hair placodes is severely affected upon sustained autonomous WNT activation. (**A**) Sagittal views and schematic of skin section depicting a basal hair bud progenitor and underlying dermal condensate (DC, encased by blue dotted lines). Labeling is for *LacZ* expression knocked into the *Axin2* locus and nuclear LEF1, two faithful proxies of WNT signaling. Additionally, nuclear SOX9 marks the overlying WNT[lo] bud cells, and P-cadherin marks the basal epithelial progenitors. Dashed lines denote the basement membrane (BM) rich in extracellular matrix (ECM) and growth factors at the epidermal-dermal border. Scale bars, 10 μm. (**B**) (top left and bottom) In utero lentiviral delivery strategy to generate sparse epidermal patches lacking APC, and therefore super-activating WNT signaling. Visual and epifluorescence imaging of mosaically transduced (R26dtTomato+) E14.5 *Apc* heterozygous and null embryos. Scale bar, 2 mm. (top right) Schematic of whole mount imaging. (**C**) Planar views of the skin surface of E14.5 embryos. Scale bar, 100 μm. (**D**) Quantifications showing *Apc* null clusters of broader size and shape than *Apc* heterozygous (het) placodes, which were analogous to wild-type in this assay (Circularity = 1 perfect circle). (Placodes and clusters density plot n > 10 mm² skin area; ****p<0.0001; Mann-Whitney test; Area and Circularity plots n = 130 placodes and 216 clusters; ****p<0.0001; Mann-Whitney test; All n ≥ 3 embryos.). (**E**). Whole mount (planar) images showing atypically strong nuclear β-Catenin and LEF1 in *Apc*-null cell clusters. Scale bar, 20 μm.

*Figure 1 continued on next page*

*Figure 1 continued*

The online version of this article includes the following source data and figure supplement(s) for figure 1:

**Source data 1.** Measurements of placodes and clusters distribution, size and shape, shown in *Figure 1D*.
**Figure supplement 1.** *Apc*-null clusters show properties of hair follicles arrested at the placode stage.
**Figure supplement 2.** *Apc*-null clusters do not present signs of DNA double strand breaks.
**Figure supplement 3.** *Apc*-null cells aggregate into clusters and are non-proliferative.
**Figure supplement 4.** *Apc*-null cell clusters lose adherens junction transmembrane protein E-Cadherin.
**Figure supplement 5.** *Apc*-null cell clusters lose the hemidesmossome integrin beta4 (ITGB4).

In contrast to wild-type and/or $Apc^{-/+}$ skin, where waves of LEF1+ placodes were patterned equidistantly in hexagonal arrays (*Zhou et al., 1995*; *Cheng et al., 2014*), hair follicle patterning was severely perturbed upon mosaic, autonomous over-stabilization of β-catenin (*Figure 1C–E*). While immunostaining revealed intense nuclear β-catenin as well as nuclear LEF1, *Apc*-null clusters were of random sizes and organization and the clusters never developed into bona fide hair buds. Instead, clusters remained uniform for the natural markers of WNT$^{hi}$ placode cells LEF1, β-catenin and Lhx2, but they failed to generate the WNT$^{lo}$ suprabasal cells that characterize the placode to hair bud transition (*Figure 1—figure supplement 1A–C*). No signs of DNA damage were observed in *Apc*-null clusters as judged by the absence of γH2AX signal (*Figure 1—figure supplement 2A–B*).

Although wild-type WNT$^{hi}$ hair bud cells are slow-cycling (*Ouspenskaia et al., 2016*), *Apc*-null clusters appeared to be altogether non-proliferative (*Figure 1—figure supplement 3A–B*). Moreover, as illuminated by co-transducing E9.5 embryos with GFP- and RFP-tagged Cre recombinase-expressing lentiviruses, both wild-type placodes and *Apc*-null clusters were multiclonal (*Figure 1—figure supplement 3C*), in agreement with the notion that WNT drives the organization of non-dividing cells into placodes within the epidermal plane (*Ahtiainen et al., 2014*). The distinct morphology of *Apc*-null clusters within the epidermis was characterized by a loss of E-cadherin but not P-cadherin (*Figure 1—figure supplement 4A–B*), suggestive of a collective cell sorting mechanism dependent on sustained WNT activation. Integrin β4 was also markedly reduced, consistent with an overall loss of polarity in these clusters (*Figure 1—figure supplement 5A–B*). Finally, the WNT$^{lo}$ (LEF1-negative) regions surrounding *Apc*-null clusters occupied a much greater than normal radius (*Figure 1C*). Taken together, our mosaic data revealed that when WNT signaling becomes too high, neighboring cells become too low for WNT signaling, sharpening the boundary between WNT$^{hi}$ and WNT$^{lo}$ cells and disrupting hair follicle patterning.

## Sustained WNT activation is characterized by a gene expression signature rich in WNT inhibitors

The results so far were suggestive of the existence of an opposing morphogen gradient within the developing skin. To search for these putative morphogenic cues, we added a fluorescent eGFP WNT-reporter to our LV-Cre lentiviral construct, so that we could use fluorescence activated cell sorting (FACS) to isolate and transcriptionally profile independent replicates of WNT-reporter$^{hi}$ and WNT-reporter$^{lo}$ cells from transduced *Apc*-null, *R26tdTomato* embryos (*Figure 2A*; *Figure 2—figure supplement 1A and B*).

When compared to their heterozygous counterparts, WNT-reporter$^{hi}$ epidermal progenitors (α6 integrin$^+$) that were null for *Apc*, displayed robust upregulation (Log2 Fold Change $\geq$ 1.5, p<0.05) of established WNT-target genes, for example *Axin2, Twist1/2* and *Bmp4*, as well as transcripts associated with WNT signaling, cell-cell signaling, cancer, epithelial-mesenchymal transition and cell adhesion (*Figure 2B–D* and *Figure 2—figure supplement 2*). Similar analysis of WNT-reporter$^{lo}$ progenitors revealed that expression of these genes was highly sensitive to cellular WNT-reporter levels, and therefore levels of WNT signaling. Intriguingly, WNT signaling sensitive genes encoded not only WNT-activators but also WNT-inhibitors, including NOTUM, WIF1, DKK4 and APCDD1.

Notably, the levels of WNT target gene expression were always higher in *Apc*-null than in non-phenotypic *Apc*-heterozygous cells, consistent with their overall ectopically higher levels of β-catenin (*Figure 2D*). Thus, it was important to verify that the WNT-sensitive genes we unearthed were relevant to normal hair follicle development.

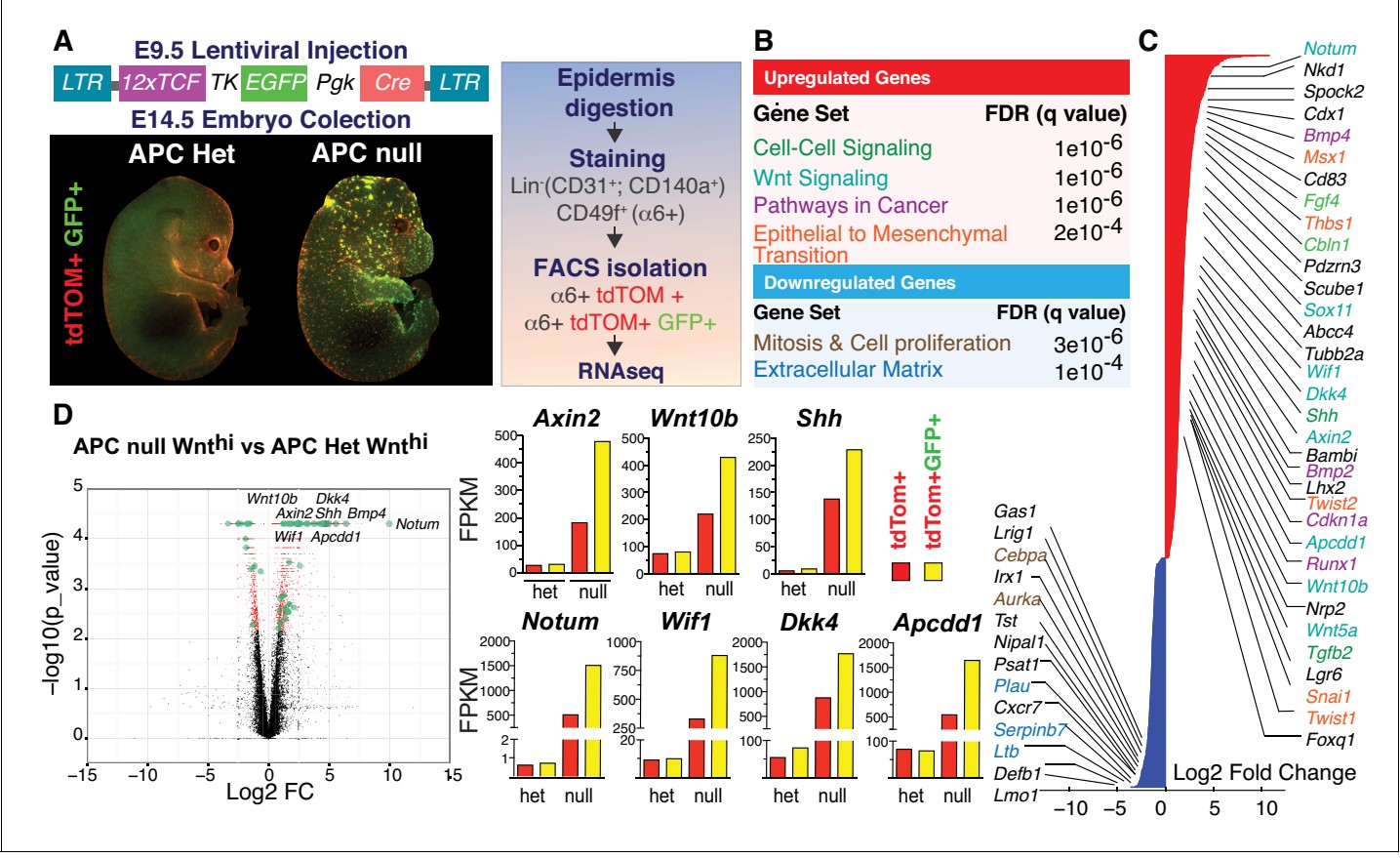

**Figure 2.** Teasing out a WNT-sensitive molecular signature based upon transcriptome profiling of skin progenitors possessing different WNT signaling levels. (**A**) LV construct, epifluorescence imaging and FACS strategy for isolating WNT signaling (GFP+) and WNT$^{lo}$ skin progenitors from LV-transduced E14.5 *Apc$^{fl/+}$* and *Apc$^{fl/fl}$; R26fl-stop-fl-tdTOM* embryos. (**B**) Gene set enrichment analysis (GSEA) of gene sets showing marked differential expression in WNT signaling progenitors from *Apc*-null vs *Apc*-het embryos. False discovery rate (FDR) q-values of enrichment are shown for each gene set. (**C**) Waterfall plot depicting genes markedly influenced (Log2 Fold Change ≥ 1.5, p<0.05) by APC status (color-coding according to B). (**D**) Volcano plot showing differentially regulated transcripts and WNT-reporter status.

The online version of this article includes the following source data and figure supplement(s) for figure 2:

**Source data 1.** Source data for the graphs shown in *Figure 2D*.
**Figure supplement 1.** FACS purification strategy to isolate WNT$^{hi}$ skin progenitors from *Apc* embryonic skin.
**Figure supplement 2.** Cell adhesion transcripts upregulated in *Apc null* WNThi cells (Geneontology – PANTHER Classification System).
**Figure supplement 2—source data 1.** Cell adhesion transcripts upregulated in *Apc null* WNThi cells (Geneontology – PANTHER Classification System).

## WNT signaling cells from developing hair follicles express high levels of WNT inhibitors

Because our lentiviral transducing strategy is not specific for placodes and the whole skin epithelium becomes transduced upon lentiviral delivery (*Beronja et al., 2010*), we devised a precise strategy to isolate a pure population of WNT$^{hi}$ signaling cells specifically from developing hair follicles. To do so, we crossed otherwise wild-type *Lhx2-GFP* and Fucci (mKO2Cdt1) mice (*Ouspenskaia et al., 2016*) and FACS-purified and profiled their slow-cycling, WNT$^{hi}$ signaling basal hair bud progenitors (α6-integrin$^{hi}$LHX2$^{hi}$mKO2$^{+}$) (*Figure 3A*; *Figure 3—figure supplement 1A*). Indeed, not only was this wild-type population WNT-reporter active (*Figure 3—figure supplement 1B*), but in addition, the transcriptome overlapped appreciably with that of the WNT$^{hi}$ potent *Apc*-null cells (*Figure 3B*; *Figure 3—figure supplement 1C*).

Most notably, WNT signaling activators and inhibitors fell within the overlap (*Figure 3—figure supplement 2*). As confirmed by in situ hybridization, many of these factors displayed expression specificity for the WNT$^{hi}$ and not WNT$^{lo}$ cells of wild-type epithelial cells (*Figure 3C*). Overall, the

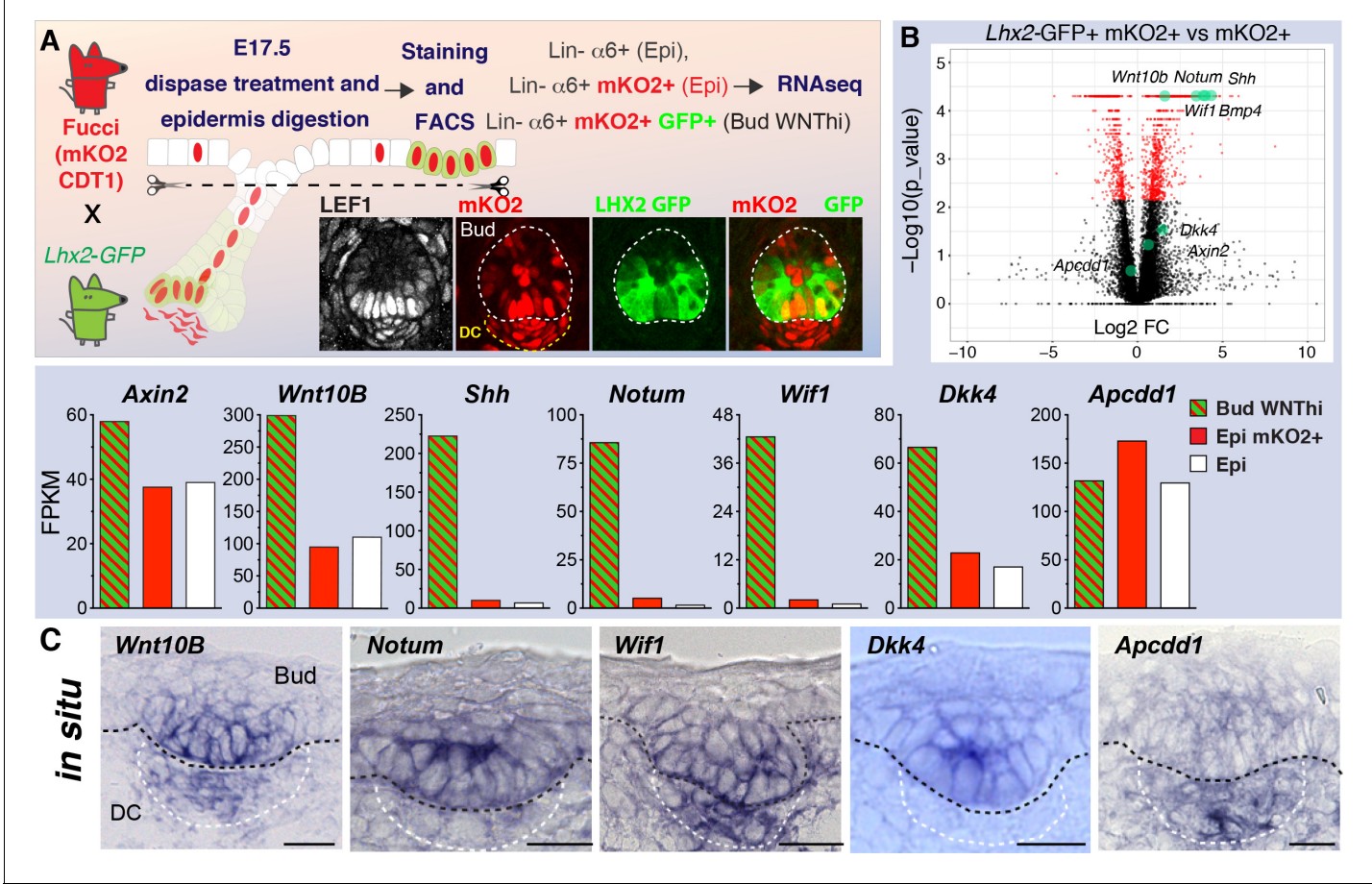

**Figure 3.** Wild-type WNT signaling progenitor cells express high levels of WNT inhibitors. (**A**) Strategy used to isolate and profile slow-cycling basal progenitors from the epidermal fraction of dispase-treated, wild-type E17.5 skin, which contains epidermis and early hair placodes/buds. Note: LEF[+] progenitors are simultaneously LHX2GFP[+] and mKO2[+]. (**B**) Volcano plot and comparative expression profiling reveals that relative to their epidermal counterparts, wild-type basal placode/bud progenitors share strong signature similarities with *Apc*-null progenitors. Green dots denote previously reported WNT target genes. (**C**) In situ hybridizations showing that WNT signaling progenitor cells simultaneously express mRNAs for WNT activators and WNT inhibitors. Black dashed lines, epidermal-dermal boundary; white dashed lines demarcate the dermal condensate (DC). Scale bars, 10 μm. The online version of this article includes the following source data and figure supplement(s) for figure 3:

**Source data 1.** Source data for the graphs shown in *Figure 3B*.
**Figure supplement 1.** FACS purification strategy to isolate WNT[hi] placode and WNT[lo] epidermal progenitors from wild-type embryonic skin.
**Figure supplement 2.** WNT[hi] signature genes in hair follicle development.
**Figure supplement 2—source data 1.** WNT[hi] signature genes in hair follicle development.
**Figure supplement 3.** BMP4 acts long range to perturb hair follicle patterning.
**Figure supplement 3—source data 1.** Source data for the graphs shown in *Figure 3—figure supplement 3A and B*.
**Figure supplement 4.** *Apc*-null cells express high levels of WNT inhibitors.

comparative analyses between WNT[hi]*Apc*-null and WNT[hi] wild-type bud cells underscored the value of comparing progenitors with different levels of WNT signaling to tease out a physiological WNT-dependent signature.

*Bmp4* was among the genes exhibiting strong WNT signaling dependency (*Figure 3—figure supplement 3A–C*). As BMPs are known to inhibit follicle formation, and BMP-inhibitors are known to promote it (*Noramly and Morgan, 1998*; *Lu et al., 2016*), this provided a possible explanation for why the hair follicle-free zone surrounding *Apc*-null clusters was increased (*Figure 1C*). Indeed, nuclear pSMAD1/5/9, a proxy for BMP4 signaling, persisted multiple cell layers away from *Apc*-null clusters (*Figure 3—figure supplement 3C*). Moreover, when over-expressed mosaically, BMP suppressed hair bud formation within adjacent regions of wild-type skin, accompanied by aberrant

expansion of pSMAD1/5/9 (*Figure 3—figure supplement 3D–G*). These findings support the notion that BMP4 acts in a long-range, negative feedback loop and is responsible for creating a bud-free environment around WNT-specified hair buds, which are driven by BMP inhibitors.

By contrast, and as previously reported for intestine (*Farin et al., 2016*), WNTs seemed to function locally, since despite marked elevation of WNT10B within *Apc*-null clusters, immunostaining did not reveal signs of long-range expansion of nuclear β-catenin/LEF1 or WNT inhibitors (NOTUM and WIF1) into surrounding wild-type skin (*Figure 3—figure supplement 4A and B*). Probing deeper into the possible functions of these counter-acting positive and negative WNT morphogens (*Langton et al., 2016*) under more physiological conditions, we continued our focus on WNT10B, NOTUM and WIF1.

## WNT ligands and WNT inhibitors are oppositely polarized by WNT signaling cells from the developing hair follicles

To track WNT inhibitors during hair follicle development, we investigated the cellular localization of their endogenous proteins at E15.5. At this time, there were three ongoing, staggered waves of hair follicle morphogenesis, enabling simultaneous capturing of placode, bud/germ and peg stages. Strikingly, WIF1 localized at the apical side of the basal cells of placode, bud and germs (*Figure 4A and B*). WIF1's apical localization in hair germs was severely impaired by Tunicamycin, suggestive of a role for N-Glycosylation in the preferential apical secretion of WNT inhibitors (*Figure 4—figure supplement 1A*; *Scheiffele et al., 1995*). By contrast, the Golgi was organized both apically and basally (*Figure 4—figure supplement 1B*).

Like WIF1, endogenous NOTUM also displayed a marked apical localization in basal bud cells (*Figure 4C*). Moreover, consistent with WIF1's role in binding and trapping WNT ligands (*Malinauskas et al., 2011*), and NOTUM's role in inactivating secreted WNTs through removal of their palmitoleate moiety (*Kakugawa et al., 2015*), nuclear LEF1 was drastically reduced in the suprabasal bud cells at the interface of this high zone of WNT-inhibitor (*Figure 4B and C*; white arrows).

To understand the importance of co-expressing quintessential WNT signaling ligands and inhibitors during hair follicle development, we devised a strategy that would allow us to similarly detect these antagonists: we exposed E9.5 *Krt14rtTA* embryos to lentiviruses harboring doxycycline-inducible expression vectors driving a C-terminal MYC-epitope tagged version of each target (*Figure 5A*). We added Doxycycline to activate *rtTA* at E13.5 and induce protein expression, and then analyzed at E15.5.

When ectopically expressed in the interfollicular epidermis, MYC-tagged WIF1 localized uniformly to epidermal cell borders, as detected by either WIF1 or MYC-tag immunofluorescence (*Figure 5—figure supplement 1A*). A similar pattern of expression was observed for MYC-tagged NOTUM (*Figure 5—figure supplement 2A*). By contrast, in the basal hair bud progenitors, both MYC-tagged inhibitors polarized apically (*Figure 5B and C*). Apical localization of WIF1-MYC and NOTUM-MYC in the developing hair follicle strikingly paralleled their endogenous localization (*Figure 4B and C*; *Figure 5B and C*; *Figure 5—figure supplement 1B*; *Figure 5—figure supplement 2B*). A similar pattern of apical expression was also observed for APCDD1 and DKK4, inhibitors that prevent WNT receptor signaling (*Figure 5—figure supplement 3A*).

By the hair germ stage, WIF1 was no longer expressed in the dermal condensate (*Figure 4B*; *Figure 5B*; *Figure 5—figure supplement 1B*), nor was NOTUM at the dermal condensate-epidermal interface (*Figure 4C*; *Figure 5C*; *Figure 5—figure supplement 2B*). Thus, by polarizing WNT-inhibitors apically in basal hair bud cells, a WNT-inhibitor free zone appeared to be generated at this epithelial-mesenchymal interface (yellow arrowheads, *Figure 4B and C*). Moreover, the robust presence of nuclear LEF1 both in basal bud cells and in the dermal condensate suggested the presence of active WNT ligands within this inhibitor-free zone. Indeed, in contrast to WNT inhibitors, WNT10B and WNT3 were both preferentially polarized at the basal membrane, as quantified by pixel intensity analyses (*Figure 5D*; *Figure 5—figure supplement 3*).

In contrast to the polarization of WNT ligands and WNT inhibitors, Frizzled-10 WNT receptor localized to all borders of the hair germ progenitor cells (*Figure 5E*). This raised the possibility that the elevation in WNT inhibitors might not be a simple negative feedback loop for WNT signaling (*Brandman and Meyer, 2008*). Rather it appeared to generate a sharp morphogen boundary, permissive for WNT signaling within basal hair bud cells and underlying dermal condensate, but

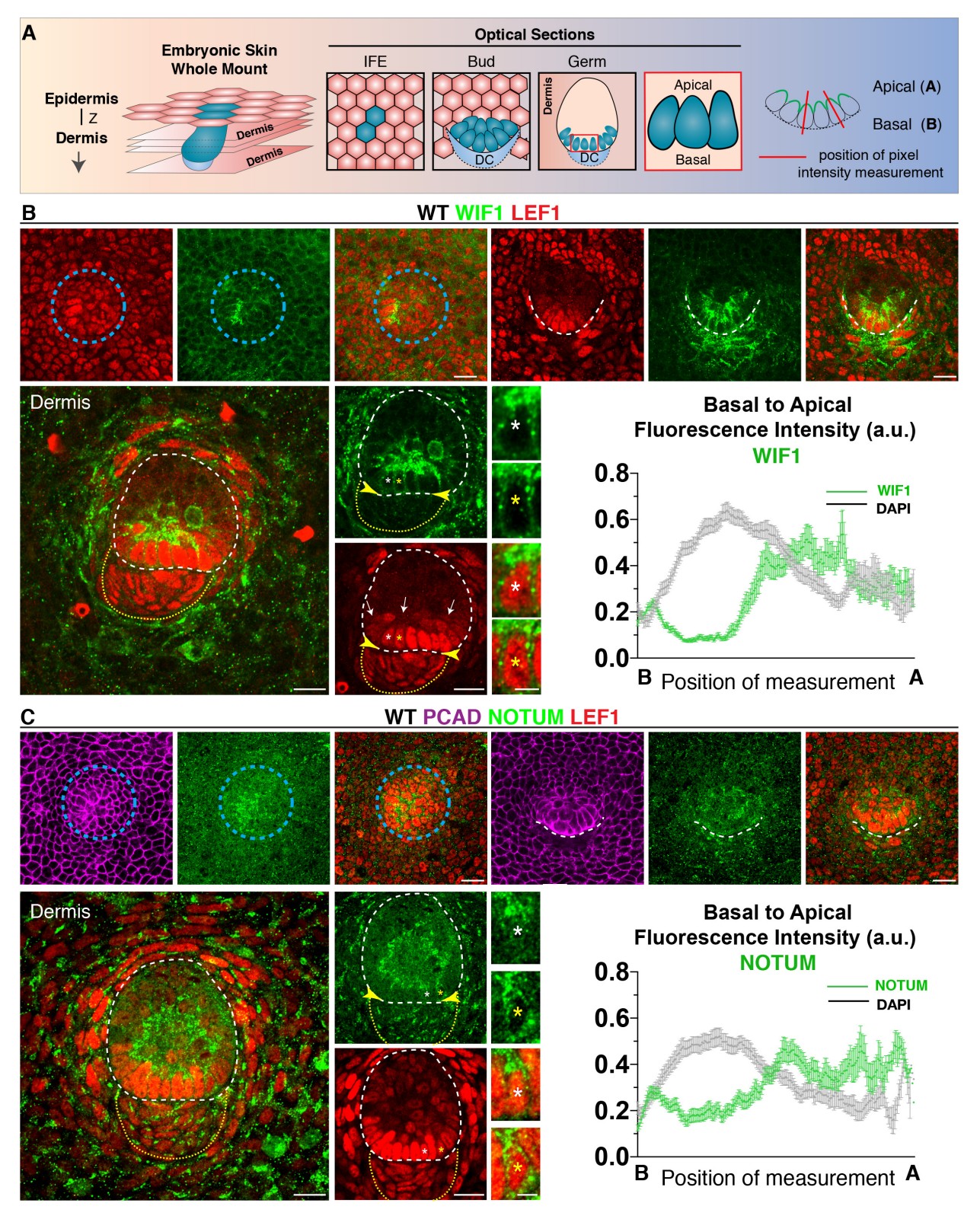

**Figure 4.** Progenitor cells apically polarize WNT inhibitors. (**A**) Schematic of whole-mount analysis from embryonic skin with examples of optical sections showing interfollicular epidermis (IFE), bud and germ; position of pixel intensity measurement. Z, plane of imaging from the Z-stack. (**B**) Anti-WIF1 and (**C**) anti-NOTUM immunofluorescence in placode, bud and germs reveal an apical accumulation of WIF1 and NOTUM. Pixel intensity profiles of basal hair bud progenitors (n = ≥40 WNT signaling progenitors; mean ± SEM; a.u., arbitrary units). Note also absence of WNT inhibitors in upper
*Figure 4 continued on next page*

*Figure 4 continued*

region of the dermal condensate (encased by yellow dotted line) at this stage of morphogenesis, leaving a WNT inhibitor free zone (yellow arrowheads) for nuclear LEF1 and WNT signaling at the epidermal-dermal boundary (white dashed line) and a WNT inhibitor high zone in suprabasal hair bud cells (arrows). Blue circular dashed lines outline placodes. White dotted lines demarcate epithelial-mesenchymal boundaries. *Denotes magnified cells, shown at right of each frame. Scale bars: 5 μm magnified cells; all others, 20 μm.

The online version of this article includes the following source data and figure supplement(s) for figure 4:

**Source data 1.** Source data for the graphs shown in *Figure 4*.
**Figure supplement 1.** Apical WIF1 localization is dependent on N-Glycosylation.
**Figure supplement 1—source data 1.** Source data for the graphs shown in *Figure 4—figure supplement 1B*.

restrictive for WNT signaling in overlying suprabasal bud cells. If so, dual expression but differential localization could explain fate diversification and morphogenesis within the developing hair follicle.

## Hair bud progenitors apically polarize WNT inhibitors to protect their own identity and differentially confer WNT signaling to their neighbors

To further probe the existence of this putative morphogen gradient across the developing hair follicles, we first devised and implemented a strategy to induce the elevation of either WIF1 or NOTUM in skin epithelial progenitors. In doing so, we observed that WNT inhibitors impaired hair follicle specification and led to a sparser hair coat (*Figure 6A*; *Figure 6—figure supplement 1A and B–B′*). We also used LGK974, which inhibits porcupine, an enzyme necessary for WNT secretion (*Liu et al., 2013*). Low doses of LGK974 administered to E15.5 skin explants were sufficient to prevent nuclear LEF1 in the normally WNT$^{hi}$ basal hair bud cells. Moreover, the normally WNT$^{hi}$ basal cells adopted the SOX9 fate of the WNT$^{lo}$ suprabasal bud cells, underscoring the importance of the WNT morphogen gradient in fate specification (*Figure 6—figure supplement 2A–C*).

At higher doses, LEF1 was lost not only from the basal hair bud cells, but also from the dermal condensates, consistent with the higher levels of nuclear LEF1/WNT signaling in the dermal condensate relative to the hair bud (*Figure 6—figure supplement 2D and E*). Moreover, when we washed out the low dose porcupine inhibitor, nuclear LEF1 and basal bud progenitor fate was restored, illustrating not only the reversibility of the process, but also the restriction of WNT signaling to the epithelial-dermal condensate boundary (*Figure 6—figure supplement 3A and B*).

The accurate and reproducible response of LEF1 expression to the porcupine inhibitor treatment and its wash-out offered yet another validation of nuclear LEF1 as a *bona fide* proxy for WNT signaling. Probing deeper, we transduced embryonic wild-type skin with our WNT-reporter and evaluated the GFP and LEF1 simultaneous expression with other WNT targets. In the basal cells from the developing hair follicle LEF1 perfectly co-localized with TCF1/7, the nuclear effector of WNT signaling. Furthermore WNT-reporter$^{hi}$, LEF1 positive progenitor cells co-expressed other key WNT signaling pathway components like FZD10 and WIF1 (*Figure 6—figure supplement 4*).

Turning to the physiological relevance of the polarized WNT inhibitors in preventing WNT signaling suprabasally, we again employed in utero lentiviral delivery, this time to transduce the embryonic skin with inducible versions of NOTUM and WIF1 that were engineered to harbor the basal targeting domain of aquaporin-4 (AQP4) (*Urra et al., 2008*; *Figure 6—figure supplement 5A*). By E15.5, transduced (H2BGFP+) hair bud progenitors displayed pronounced basal targeting of these AQP4-tagged WNT-inhibitors (*Figure 6B*; *Figure 6—figure supplement 5B–E*). Quantifications showed that basal targeting was more efficient with NOTUM, and this correlated with a more pronounced reduction in hair follicles. Moreover, as quantified by nuclear LEF1 fluorescence intensity, the nuclear LEF1 signal was significantly decreased in NOTUM-AQP4-induced cells compared to either NOTUM-induced or wild-type cells (*Figure 6C*). This was particularly clear in mosaic hair buds, where basal progenitors that did not express NOTUM-AQP4-MYC-tag (arrowheads, *Figure 6C*) were adjacent to their NOTUM-AQP4-transduced counterparts. These manipulations also resulted in an expansion of WNT$^{lo}$ SOX9 cells, underlying the importance of properly regulating WNT inhibitors in hair follicle morphogenesis (*Figure 6D*).

Finally, we tested the functional importance of NOTUM's apical localization by asking whether its depletion would lead to an increase in WNT signaling. By transducing *Notum*$^{fl/fl}$ and *Notum*$^{fl/+}$ *R26td Tomato* embryos with LV-Cre, we found that in the absence of NOTUM, both the proportion

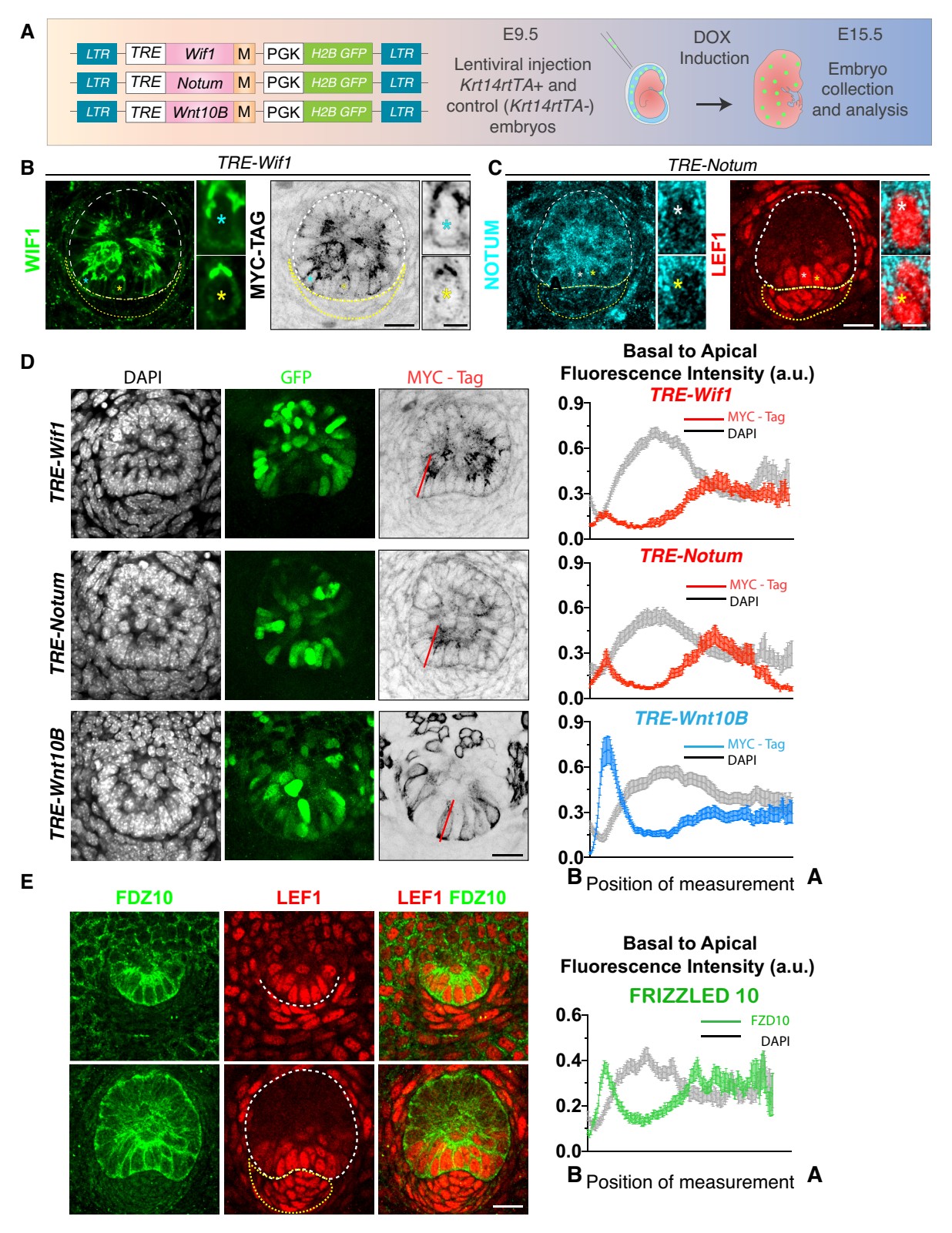

**Figure 5.** Evidence of oppositely polarized and short-range action of WNTs and WNT inhibitors in hair bud progenitors that are actively signaling through WNTs. (**A**) LV-constructs and strategy to monitor WNT inhibitors and WNTs. M, myc-tag. TRE, tetracycline regulatory element. *Krt14rtTA* is a transgenic mouse line expressing a doxycycline (DOX)-inducible transcriptional activator for TRE. (**B, C**) Similar to endogenous expression, anti-WIF1 and anti-NOTUM immunofluorescence on MYC-tag transduced skin show apical localization in hair bud progenitors. (**D**) Anti-MYC-tag

*Figure 5 continued on next page*

*Figure 5 continued*

immunofluorescence of transduced skins revealing apical polarization of NOTUM and WIF1, but basal polarization of WNT10B. At right are basal-apical MYC-Tag/DAPI pixel intensity profiles of basal hair bud progenitors (n = ≥40 WNT signaling progenitors; mean ± SEM; a.u., arbitrary units). (E) Pixel intensity profile and immunolocalization of endogenous WNT-receptor FRIZZLED 10 shows uniform localization at borders of hair bud and germ WNT signaling cells (n = 37 cells; mean ± SEM; a.u., arbitrary units). *Denotes magnified cells, shown at right of each frame. Scale bars: 5 µm magnified cells; all the others, 20 µm.

The online version of this article includes the following source data and figure supplement(s) for figure 5:

**Source data 1.** Source data for the graphs shown in *Figure 5D and E*.
**Figure supplement 1.** WIF1 expression during early hair follicle development.
**Figure supplement 2.** NOTUM expression during early hair follicle development.
**Figure supplement 3.** Differential polarization of WNT inhibitors and activators.
**Figure supplement 3—source data 1.** Source data for the graphs shown in *Figure 5—figure supplement 3A*.

of LEF1+ cells and also their nuclear LEF1 signal intensity were significantly increased within developing hair buds (*Figure 7A–C*). Moreover, the effects of *Notum* ablation were largely confined to the apical region of the hair bud and not the underlying dermal condensate, further underscoring the short range and functional importance of apically localizing WNT inhibitors.

## Discussion

Pattern formation plays near universal roles in tissue morphogenesis. The early developing skin is composed of a single layer of multipotent epithelial progenitor cells that will either stratify and develop into the skin's epidermal barrier or form epithelial placodes to launch hair follicle morphogenesis. Positional cues are important not only to specify the uniform distribution of hair follicles across the tissue, but also to differentiate the cells within each of these mini-organs. In uncovering the existence of an internal WNT morphogen gradient within the earliest progenitors of the hair placode, we have begun to understand how WNT signals can be directionally distributed to neighboring cells to break symmetry and trigger the morphogenetic transition from the two-dimensional early placode to a three-dimensional mini-organ.

WNT signaling has long been known to be important broadly for regenerative and morphogenetic processes (*Petersen and Reddien, 2011*; *Loh et al., 2016*; *Clevers et al., 2014*). The presence of inhibitors of the WNT signaling pathway has also long been recognized, and given the oft short-lived nature of WNT signals in development, it has always been assumed that inhibitors function in a negative feedback loop to turn off the signal for the next step in lineage specification. Although the existence of such feedback loops is well-established (*Perrimon and McMahon, 1999*), such a mechanism did not reconcile how WNT signaling remains high in basal hair bud progenitors that also simultaneously express at least four different WNT inhibitors. An additional conundrum was how this WNT expressing, WNT signaling progenitor gives rise to only one daughter cell that retains this status, while the other daughter cell adopts a WNT-restricted state.

Our findings show that by differentially compartmentalizing WNTs and WNT inhibitors, basal placode progenitors not only maintain both positive and negative WNT morphogens simultaneously, but also directionally target the signals, providing the requisite positional cues to transition from two to three dimensional patterning within the developing tissue. By polarizing WNTs basally, progenitors are able to retain their own WNT signaling as well as that of their underlying mesenchymal neighbors to fuel hair follicle morphogenesis at the dermal-epidermal interface. Conversely, by polarizing WNT inhibitors apically, the same hair bud progenitors directionally orchestrate WNT$^{lo}$SOX9+ fate specification of their overlying neighbors to launch the diversification of the epithelial cells within the developing hair follicles (*Figure 8*).

During development, the formation of precise boundaries is fundamental to the specification of different cellular compartments. Our findings best fit a model whereby developing hair follicle progenitors use WNTs and WNT inhibitors to build local boundaries. By localizing WNT inhibitors apically, WNT$^{high}$ progenitors limit the WNT response, preserving their own WNT$^{high}$ signaling identity while simultaneously preventing their suprabasal daughter from responding to the WNT signal.

A similar refining mechanism has been previously proposed in the *Drosophila* imaginal disks. In the *wingless* expressing cells of the wing margin, cells that are closer to the dorsoventral boundary

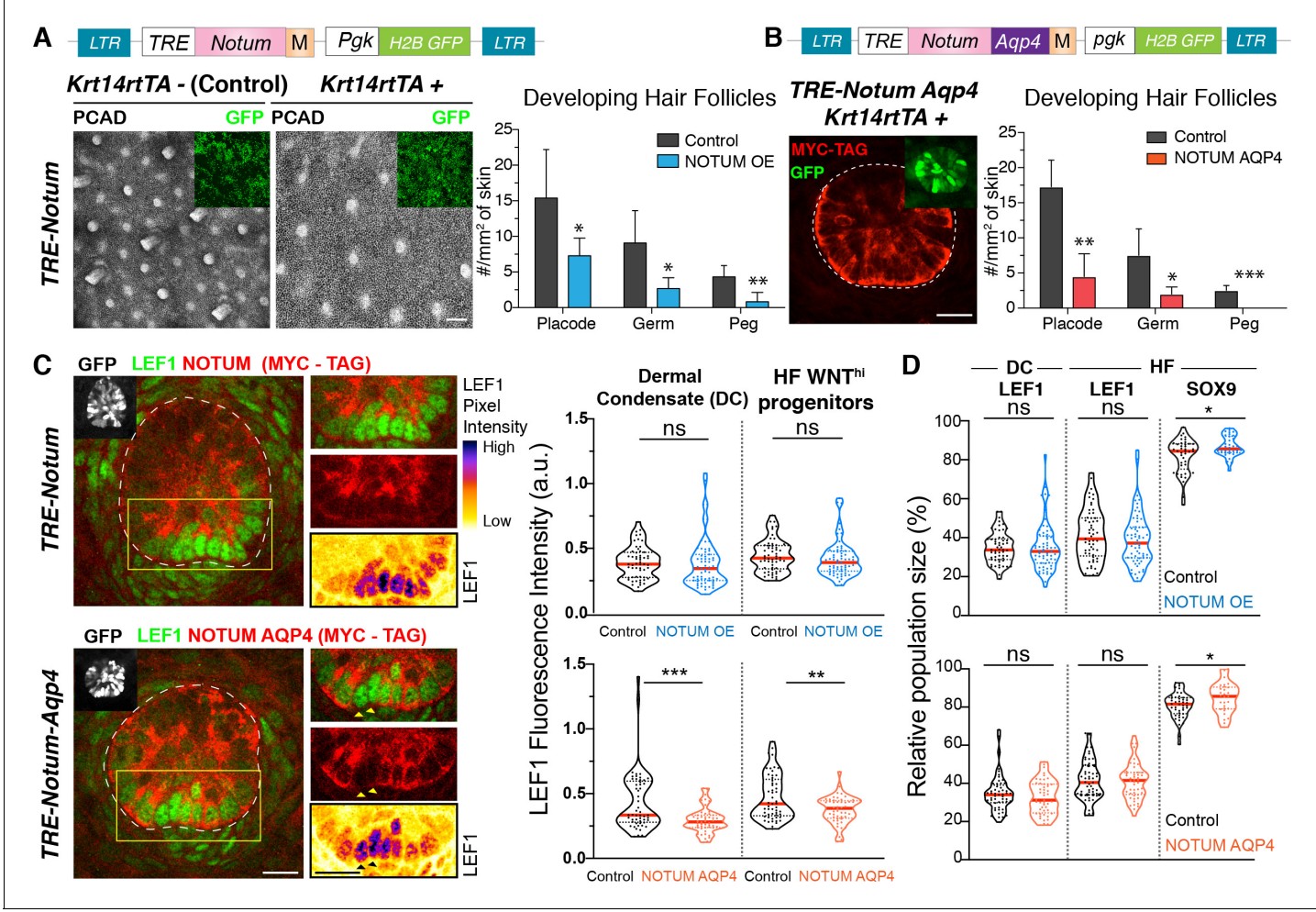

**Figure 6.** Hair bud progenitors apically polarize WNT inhibitors to protect their own identity and differentially confer WNT signaling to their neighbors. (A) Whole-mount immunofluorescence and quantifications reveal that elevating NOTUM across the epidermal plane results in significantly fewer hair follicles (Mean ± SD; n > 10 mm$^2$ skin analyzed from ≥3 embryos; *p<0.05; **p<0.005; Mann-Whitney test). Scale Bar, 100 μm. OE, overexpression. Insets verify transduced regions. All scale bars for immunofluorescence images are 20 μm. (B) Adding an aquaporin4-tag mispolarizes NOTUM to the basal side of hair bud progenitors. Quantifications reveal that mis-polarizing a WNT inhibitor poses a significant impediment to hair follicle morphogenesis (Mean ± SD; n = 8 mm$^2$ skin analyzed from ≥3 embryos; *p<0.05; **p<0.005; ***p<0.0005; unpaired Student t test). White dotted lines demarcate epithelial-mesenchymal borders throughout. (C) Whole-mount immunofluorescence and quantifications of normalized LEF1 pixel intensities reveals that NOTUM mis-polarization leads to a significant decrease of LEF1 intensity in WNT signaling cells from both the dermal condensate and the hair bud (n ≥ 48 hair follicles from ≥3 embryos each; Mann-Whitney test ***p=0.0002 and unpaired t test **p<0.005; n.s. non-significant; red lines represent the distributions' median). Yellow boxes show regions magnified at right. Arrowheads show two cells not expressing NOTUM-AQP4-MYC-Tag, which have higher LEF1 signal than their expressing neighbors. (D) Violin Plots show that increasing levels of NOTUM and NOTUM-AQP4 lead to an increase of SOX9 expressing cells (n ≥ 30 developing hair follicle from at least three different embryos; Mann-Whitney test TRE-NOTUM *p=0.0389 and TRE-NOTUM-AQP4 *p=0.0461 n.s. non-significant; red lines represent the distributions' median).

The online version of this article includes the following source data and figure supplement(s) for figure 6:

**Source data 1.** Source data for the graphs shown in *Figure 6A, B, C, and D*.

**Figure supplement 1.** WNT inhibitor overexpression perturbs hair follicle formation.

**Figure supplement 1—source data 1.** Source data for the graphs shown in *Figure 6—figure supplement 1A*.

**Figure supplement 2.** Hair bud progenitors cannot maintain their fate upon WNT inhibition.

**Figure supplement 2—source data 1.** Source data for the graphs shown in *Figure 6—figure supplement 2A, C, and E*.

**Figure supplement 3.** Hair bud progenitors revert their fate after WNT inhibitor washout.

**Figure supplement 3—source data 1.** Source data for the graphs shown in *Figure 6—figure supplement 3A*.

**Figure supplement 4.** Hair follicle progenitor cells co-express WNT reporter, LEF1, TCF1/7 and WNT target gene products FZD10 and WIF1.

**Figure supplement 5.** NOTUM and WIF1 mis-localization leads to impaired development of hair follicles.

**Figure supplement 5—source data 1.** Source data for the graphs shown in *Figure 6—figure supplement 5B, C, and E*.

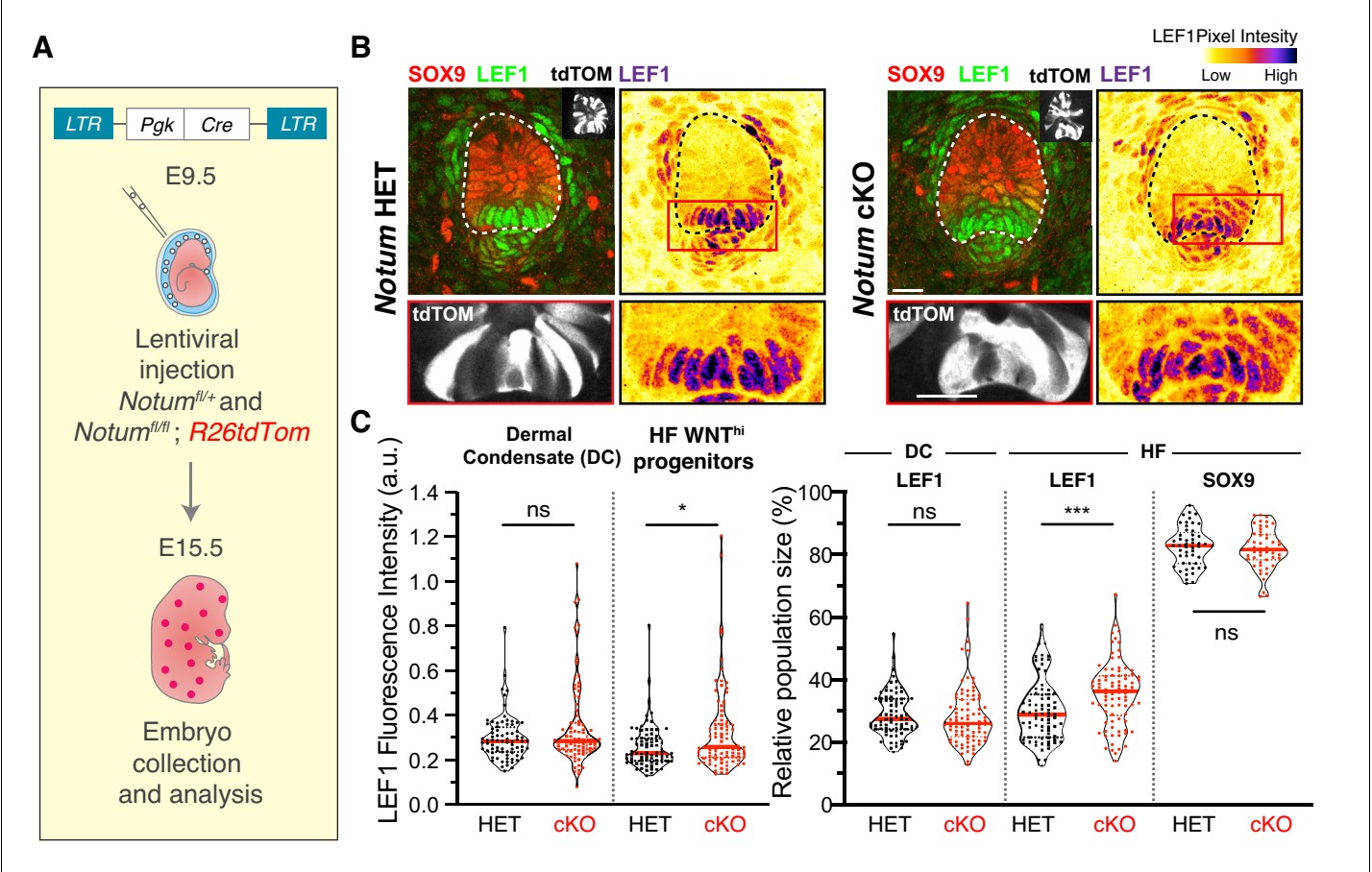

**Figure 7.** Notum regulates the formation of sharp boundaries between neighboring cell fates. (**A**) In utero lentiviral delivery strategy to conditionally ablate *Notum* in *R26dtTomato* embryos. (**B–C**) Representative whole-mount immunofluorescence showing LEF1 intensity profile and population size in *Notum* -/+ and *Notum* -/- skin. Red boxes denote regions magnified below each image. Note that *Notum* ablation (but not heterozygous) leads to an increase in LEF1 signal and LEF1+ cell populations in the WNT signaling progenitor cells (n ≥ 75 hair follicles from ≥3 different litters analyzed; Mann-Whitney test *p=0.0332 and ***p=0.0002; n.s. non-significant; red lines represent the distributions' median).
The online version of this article includes the following source data for figure 7:

**Source data 1.** Source data for the graphs shown in *Figure 7C*.

are able to repress *wingless* expression in their juxtaposed neighbors through a self-refining mechanism. In this case, however, the mechanism appears to involve NOTCH, whose activity is required for *wingless* expression, which in turn appears to repress NOTCH activity (*Rulifson et al., 1996*). In the hair follicle, NOTCH signaling has not been detected in the WNT10B+ progenitors, but rather the WNT[low] differentiating cells (*Blanpain et al., 2006*). Thus, while the mechanisms seem to be evolutionarily divergent, the functional output is similar and involves the establishment of a sharp boundary that enables the emergence of juxtaposed cell fates.

Although our current study focused on the existence of this single-cell length morphogen gradient and its functional significance, it will be interesting in the future to unravel how bidirectional targeting occurs. WNTs are known to be N-glycosylated, and studies in *Drosophila* suggest that WNTs can be secreted apically and then transported basally (*Yamazaki et al., 2016*). Intriguingly, however, in *Drosophila*, N-glycosylation-deficient Wingless is secreted without consequence (*Tang et al., 2012*), and in mammalian cells, WNTs have been found to be more potent and bind extracellular matrix more robustly in the absence of N-glycosylation (*Doubravska et al., 2011*). Thus, although our tunicamycin results suggest a role for N-glycosylation in WNT-inhibitor apical secretion, it may be advantageous for cells such as hair bud progenitors that adhere to a basement membrane to secrete their WNTs basally. The ability of ECM to retain growth factors (*Baeg et al., 2001*), including WNT regulators, makes this hypothesis all the more attractive.

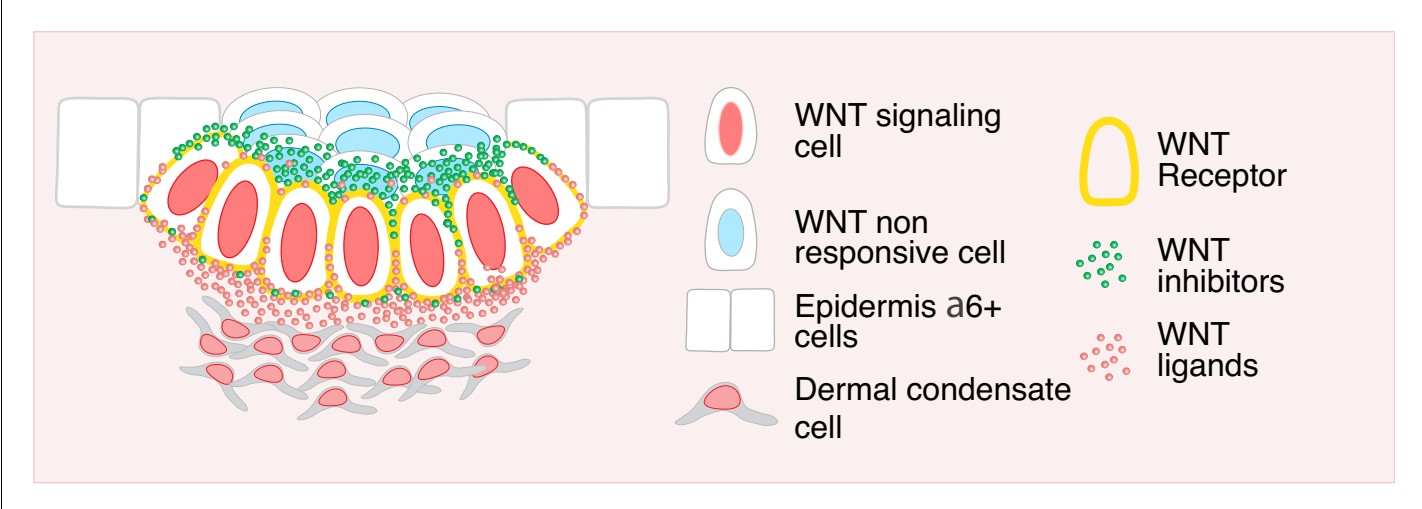

**Figure 8.** Summarizing model. WNT signaling basal progenitors form opposing intracellular morphogen gradients of WNT inhibitors and WNT ligands/activators. In so doing, they preserve their own WNT signaling and identity and directionally permit (dermal condensate) or restrict (suprabasal hair bud cells) WNT signaling in surrounding neighbors.

In closing, by establishing a morphogen gradient at the cellular level, signals are constrained such that two neighboring cell populations in direct physical contact can effectively receive different signaling inputs. Overall, the ability to directionally control rapid changes in daughter fates, and to establish sharp tissue boundaries without the need for direct competition and/or elimination, offers basic advantages to this mechanism that are likely to be broadly applicable in development.

## Materials and methods

### Mouse strains, lentiviral transduction and constructs

Mice were housed and cared for in an AAALAC-accredited facility at the Rockefeller University. All animal experiments were conducted in accordance with protocols approved by IACUC and in accordance with National Institutes of Health guidelines. All animal procedures used in this study are described in our #17020-H protocol named *Development and Differentiation in the Skin*, which had been previously reviewed and approved by the Rockefeller University Institutional Animal Care and Use Committee (IACUC). All animals used for the experiments in this manuscript were generated previously: *Axin2-LacZ* (The Jackson Laboratory) (*Lustig et al., 2002*), *Krt14-rtTA* (*Nguyen et al., 2006*), *Rosa26Flox-Stop-Flox-tdTom* (The Jackson Laboratory), *Apcfl/fl* was a kind gift from Kucher-lapati Lab (*Kuraguchi et al., 2006*), *Fucci* (595, Riken,*Sakaue-Sawano et al., 2008*), *Lhx2-EGFP* (The Gene Expression Nervous System Atlas (GENSAT) Project, NINDS Contracts N01NS02331 and HHSN271200723701C to The Rockefeller University, New York, NY, USA), *Wif1-KO* was a kind gift from Igor Dawid (NIH), *Notum-KO* embryos and *Notumfl/fl* mice were kind gifts from the Jean Paul-Vincent Lab (*Canal et al., 2016*).

We used ultrasound-guided in utero lentiviral-(LV) mediated delivery of RNAi into the amniotic cavity of living E9.5 mouse embryos. This non-invasive technique selectively infects and transduces the single-layer of unspecified epidermal progenitors as previously described (*Beronja et al., 2010*). The construct for lentiviral *Pgk-NLS-Cre-mRFP* has been described (*Williams et al., 2011*). *Pgk-NLS-Cre-EGFP* was generated by replacing the mRFP coding region with EGFP. For our LV-WNT reporter (*pLKO-TK-12xTOP-EGFP-Pgk-Cre*), *Cre* was amplified by polymerase chain reaction (PCR) from *pLKO.1-Pgk-Cre* (*Williams et al., 2011*) and inserted into pLenti-12xTOP-EGFP, in which EGFP is driven by a minimal herpes virus thymidine kinase promoter downstream of an enhancer containing multimerized LEF1 DNA binding sites (*Beronja et al., 2013*). Lentiviral doxycycline-inducible constructs were cloned using a tetracycline regulatory element (TRE) sensitive to the binding and activation by the doxycycline-inducible rtTA transcription factor. This TRE system (*LV-TRE-Gene-Pgk-*

H2BGFP) has been previously described (*Hsu et al., 2014*). The cDNAs, *Bmp4* (MG50439-G, Sino Biological, *Lu et al., 2016*), *Notum-Myc-tagged* (MR217230, Origene), *WIF1-Myc-tagged*, (MR202510, Origen) *Wnt10b-Myc-tagged* (MR224739, Origen), *Dkk4-Myc-tagged* (MR202533, Origene), *Apcdd1-Myc-tagged* (MR225129, Origene), *Wnt3-Myc-tagged* (MR222492, Origene) were purchased from Origen, and then cloned by PCR to insert the gene of interest (GOI) in the *LV-TRE-GOI-Pgk-H2BGFP*. *Notum-Aqp4-Myc*-tagged, and *Wif1-Aqp4-Myc*-tagged were designed by adding the coding sequence of the last 42 amino acids of the rat Aquaporin-4 (*Madrid et al., 2001*; *Urra et al., 2008*), upstream of *Myc-tag* and synthesized by Genewiz. *Notum-Aqp4-Myc-tagged* and *Wif1-Aqp4-Myc-tagged* further cloned by PCR and inserted into the *LV-TRE-GOI-Pgk-H2BGFP*. *Krt14rtTA* was activated by feeding pregnant females with doxycycline (2 mg/kg, Doxy-feed, Bio-Serv) chow at E9.5 until time of collection.

## Flow cytometry

Methods for preparing embryonic mouse back and head skin for fluorescence activated cell sorting (FACS) and purification of α6-high epidermal and hair bud progenitors have been previously described (*Williams et al., 2011*). Briefly, the skin of E14.5 and E17.5 embryos was dissected and either (E14.5) placed directly into a trypsin-EDTA solution at 37°C for 5 min on an orbital shaker, or (E17.5) first treated with the enzyme dispase (Gibco, 1:1 in PBS) overnight at 4°C prior to making the single cell suspension. Sorting buffer (PBS 5% FBS) was added to the suspension to neutralize trypsin. Single-cell suspensions were obtained by filtering through a 70 μM strainer and collected by centrifugation at 300 g for 5 min. Cell suspensions were washed three times and incubated with the appropriate antibodies for 30 min on ice. For FACS, we used the following antibodies (along with epifluorescent markers): α6-integrin (eBiosciences) to select for basal progenitors, CD140a (PDGFRA) (eBiosciences) to select against mesenchymal cells, CD31 (PECAM1) (eBiosciences) to select against platelets. DAPI was used to exclude dead cells. Cell isolations were performed on FACS Aria sorters running FACS Diva software (BD Biosciences).

## RNA-seq and analysis

FACS isolated keratinocytes, pooled from three embryos for each condition, were sorted directly into TrizolLS (Invitrogen). RNA was purified using Direct-zol RNA MiniPrep kit (Zymo Research) per manufacturer's instructions and 2-pooled samples were sequenced for each condition. The quality of the RNA for sequencing was determined using an Agilent 2100 Bioanalyzer and all samples analyzed had RNA integrity numbers (RIN) >8. Library preparation was performed by the Weill Cornell Medical College Genomic Core facility, which uses the Illumina TrueSeq mRNA sample preparation kit. RNAs were sequenced on their Illumina HiSeq 2500 machines. The reads were aligned with Tophat using mouse genome build mm9 build and the transcript assembly and differential expression was performed using Cufflinks with Ensembl mRNAs to guide assembly. Analysis of RNA-seq data was done using the cummeRbund package in R (*Trapnell et al., 2012*).

The genes known to be sensitive to WNT signaling (http://web.stanford.edu/group/nusselab/cgi-bin/wnt/) are marked as green dots in the volcano plots that compare WNT$^{hi}$ and WNT$^{lo}$ transcriptomes of embryonic skin progenitors on *Apc*-null and *Apc*-het mice. Selected genes relevant for this study are highlighted in both volcano plots (WNT$^{hi}$ and WNT$^{lo}$ transcriptomes of embryonic skin progenitors on *Apc*-null and *Apc*-het and in WT backgrounds - Lhx2GFP+ mKO2Cdt1+ vs mKO2Cdt1 +). Differentially regulated transcripts were analyzed with Gene Set Enrichment Analysis (GSEA) to find enriched gene sets (*Subramanian et al., 2005*).

The overlap between *Apc*-null WNT$^{hi}$ and wild-type WNT$^{hi}$ (Lhx2GFP+ mKO2Cdt1+) signature genes was defined by intersecting significantly differentially expressed genes (those with a q-value of <0.05 and with Log2 fold change -FC- of 1.5 fold up) in the two populations. The significance of the overlap was evaluated with a *P-value* derived using the hypergeometric distribution using R software.

## In situ hybridization

Two different protocols were used to perform in situ hybridization depending on the probes hybridized. Whereas protocol one was used for *Wnt10b*, *Apcdd1* and *Dkk4* hybridization, protocol two was used for the *Notum* and *Wif1* hybridizations. The Wnt10b anti-sense probe was synthesized

using the cDNA region 1493–2008 bp from the mRNA annotated as NM_011718.2 (PCRII-*Wnt10b*). The cDNA was linearized with the restriction enzyme XhoI and transcribed with Sp6 polymerase. The cDNA used to synthesize the *Apcdd1* anti-sense probe (pCR4-*mApcdd1*) was a generous gift from Angela Cristiano. The cDNA was linearized with the restriction enzyme SpeI and transcribed with T3 polymerase. The cDNA used to synthesize *Dkk4* anti-sense probe (pGEMT-*mDkk4*) was a generous gift from David Schlessinger. The cDNA was linearized using the restriction enzyme NcoI and was further transcribed with the SP6 polymerase. The *Notum* anti-sense probe was synthesized using the cDNA region 385–1495 bp from the mRNA annotated as NM_175263.4. The cDNA was linearized with NotI and transcribed with SP6 polymerase. The *Wif1* anti-sense probe was synthesized using the cDNA region 1289–2037 bp from the mRNA annotated as NM 011915.2 (pCRII-*mWif1_3*). The cDNA was linearized with NotI and transcribed with SP6 polymerase.

## Protocol 1
10 to 14 µm cryosections were fixed for 10 min in 4% paraformaldehyde (PFA, from 16% PFA solution Electron Microscopy Sciences) in Diethyl pyrocarbonate-PBS (DEPC-PBS), and washed with DEPC-PBS (two times, 5 min each). Sections were incubated in TEA buffer with 0.25% acetic anhydride (10 min) and washed with DEPC-PBS (three times, 5 min each). Pre-hybridization of tissue sections was performed at 68°C for 2 hr with hybridization buffer (50% deionized formamide, 5X saline-sodium citrate, SCC, 0.5 mg/ml salmon sperm DNA, 0.5 mg/ml yeast tRNA and 8.5X Denhardt's solution). Hybridization with 1 ug/ml of probe was preformed overnight at 68°C (for 18 hr). To remove the unbound probe, sequential stringent washes were performed at 68°C (5 min with 5X SSC, followed by three times 30 min with 0.2X SSC), and at room temperature (RT, 5 min with 0.2X SSC followed by 10 min with B1 buffer - 100 mM Tris-HCl pH 7.5, 0.15 M NaCl). Tissue was blocked with 10% Normal Goat Serum (NGS) in B1 buffer (1 hr at RT) before Digoxigenin detection. Sections were incubated overnight at 4°C with Anti-Digoxigenin-AP, Fab fragments (from sheep, Roche, 1:2000 in B1 buffer with 1% NGS) and washed with B1 buffer (three times, 10 min each). Finally, slides were protected from light and developed at RT with BM purple containing 0.24 mg/ml levamisole and 0.1% Tween-20 until satisfactory signal was achieved.

## Protocol 2
10 to 14 µm cryosections were prepared one day prior to the procedure, stored at –80° C or kept on dry ice until ready for fixation. Sections were fixed for 30 min in cold (4% PFA at 4° C) and washed three times (5 min each) at RT with DEPC-PBS. Slides were treated with 3% $H_2O_2$ (30 min) and washed with DEPC-PBS (three times, 5 min each). Slides were equilibrated with TEA buffer (5 min), treated with TEA buffer containing 0.25% acetic anhydride (10 min), and washed three times with DEPC-PBS (5 min each). Tissue pre-hybridizion was performed for 2 hr at 68°C with hybridization buffer (50% deionized formamide, 2X SSC, 10% dextran sulfate, 0.5 mg/ml yeast tRNA, 0.5 mg/ml heat-denatured salmon sperm DNA) and hybridization was performed with 1 µg/ml of probe overnight at 68°C (for 18 hr). Post-hybridization washes were performed at 68°C (10 min with 5X SSC, and three times 30 min with 0.2X SSC). Slides were then washed at RT with 0.2X SSC (for 5 min) before incubation with blocking solution for 1 hr at RT (0.5% Roche Blocking reagent in B1 buffer, 100 mM Tris-HCl pH 7.5, 0.15 M NaCl). A second block was performed (1 hr at RT) with B1-BTx buffer (100 mM Tris-HCl pH 7.5, 0.15 M NaCl, 1% BSA, 0.3% Triton-X 100). Sections were incubated overnight at 4° C with anti-Digoxigenin-AP, Fab fragments (1:2000 in B1-BTx buffer), and washed at RT with sequential washes; 1) four times, 20 min each, with B1-BTx buffer; 2) 5 min with B1 buffer and finally 3) 5 min with B3 buffer (100 mM Tris-HCl pH 9.5, 0.1 M NaCl, 50 mM MgCl2). Signal was developed protected from light and incubating sections with BM purple (with 0.24 mg/ml levamisole, 0.1% Tween-20).

## Whole-mount immunofluorescence and histological analyses
5-ethynyl-2′-deoxyuridine (EdU, 500 µg/g, Life Technologies) was injected intraperitonally into pregnant females 4 hr prior to processing embryos at the desired stage of development. Typically >3 embryos from independent experiments were analyzed per condition. For whole-mount immunofluorescence, embryos were fixed in 4% PFA in phosphate buffered saline (PBS) for one hour, followed by extensive washing in PBS. Samples were then permeabilized for 3 hr in 0.3% Triton X-100 in PBS

and treated with Gelatin Block (2.5% fish gelatin, 5% normal donkey serum, 3% BSA, 0.3% Triton, 1X PBS). For immunolabeling with mouse antibodies, sections were first incubated with the M.O.M. blocking kit according to manufacturer's instructions (Vector Laboratories). The following primary antibodies were used: P-Cadherin (goat, 1:300; R and D AF761), LEF1 (guinea pig, 1:2000 and rabbit 1:300, Fuchs Lab; rabbit, 1:300, Cell Signaling C12A5), SOX9 (guinea pig, 1:2000; Fuchs Lab), LHX2 (rabbit, 1:2000; Fuchs Lab), anti-GFP/YFP (chicken, 1:1200; Abcam), anti-RFP (rat 1:200; Chromotek 5F8) β-catenin (mouse, 1:200, BD 610154), pSMAD 1/5/9 (rabbit, 1:200; Cell Signaling), NOTUM (rabbit 1:100; Sigma HPA023041), WIF1 (goat 1:300; R and D AF7135), MYC-tag (rabbit, 1:300; Cell Signaling 71D10), SOX2 (rabbit, 1:200; Abcam EPR3131), Trans-Golgi (rabbit 1:200; abcam TGN46 16059), Frizzled10 (rabbit 1:200; MyBioSource MBS9606335), SHH (goat 1:50; R and D AF445), E-Cadherin (rabbit 1:500; Cell Signaling, 24E10), P-Histone H2AX S139 (rabbit 1:200; Cell Signaling), β4-Integrin (rat 1:500; CD104 346-11A BD), TCF1/TCF7 (rabbit 1:500; Cell Signaling C63D9). Primary antibodies were incubated at 4°C for 36 hr. After washing with 0.1% Triton X-100 in PBS, samples were incubated overnight at 4°C with secondary antibodies conjugated with Alexa 488, RRX, or 647 (respectively, 1:1000, 1:500, and 1:200, Life Technologies). Samples were washed, counterstained with 4'6'-diamidino-2-phenilindole (DAPI) and mounted in SlowFade Diamond Antifade Mountant (Invitrogen), and EdU incorporation was detected by Click-It EdU AlexaFluor 647 Imaging Kit (Life Technologies).

## Immunohistochemistry and LacZ-derived β-galactosidase activity

For sagittal analyses of tissues, pre-fixed (4% PFA in PBS), paraffin-embedded embryos were sectioned at 10 µm. Immunohistochemistry was performed by incubating sections at 4°C overnight with primary antibodies against mouse anti-β-catenin (mouse, 1:1000; Sigma, 15B8) and APC (rabbit 1:500; Sigma HPA013349). For brightfield immunohistochemistry, biotinylated species-specific secondary antibodies followed by detection using (ImmPRESS reagent kit peroxidase Universal - Vector Laboratories) and DAB kit (ImmPACT DAB Peroxidase (HRP) Substrate Vector Laboratories) were used according to the manufacturer's instructions.

LacZ-derived β-galactosidase activity was assayed on frozen sections (10 µm), fixed with 0.5% glutaraldehyde in PBS for 2 min, washed with PBS, and then incubated with 1 mg/ml Xgal substrates in PBS with 1.3 mM $MgCl_2$, 3 mM K3Fe(CN)6, and 3 mM K4Fe(CN)6 for 1 hr at 37°C.

## Skin explants and pharmacological treatment

Head and back skins were excised from E15.5 embryos and placed into sterile PBS. Explants were cut in half to compare morphogenesis of pharmacologically-treated *vs* vehicle control skin. Each explant half was covered with Nucleopore TrackEtch filters (Whatman) dermis side down. Filters with skin samples were placed in lummox teflon-bottom dishes (Sarstedt). Pre-warmed keratinocyte culture medium with 0.3 mM calcium was added to the culture. Each corresponding half skin received one treatment: either Tunicamycin (0.15 mM, 1 mM and 2 mM; Milipore Sigma) or DMSO control. Explants were cultured at 37°C, 5% CO2 for 10 hr and fixed with 4% PFA for 45 min before immunostaining and confocal microscopy analysis.

For the porcupine inhibitor experiment, each half skin was treated with either porcupine inhibitor LGK974 (1 mM or 10 mM; Cayman Chemical), or DMSO control. Explants were kept at 37°C, 5% CO2. Media with treatment was changed after 12 hr. After culturing for 24 hr or 36 hr at 37°C, samples were fixed and processed for confocal immunofluorescence microscopy.

Porcupine inhibitor washout experiment was performed by treating each half skin with 1 mM LGK974. After 12 hr one of the samples was fixed (PFA 4%) while the media was changed every 5 min (total of 20 min) for the corresponding other sample. Washout sample was kept at 37°C, 5% CO2 for additional 24 hr and fixed with 4% PFA before immunostaining and confocal microscopy analysis.

## Confocal microscopy

Confocal images were acquired using a spinning disk confocal system (Andor Technology Ltd) equipped with an Andor Zyla 4.2 and a Yokogawa CSU-W1 (Yokogawa Electric, Tokyo) unit based on a Nikon TE2000-E inverted microscope. Four laser lines (405, 488, 561 and 625 nm) were used for near simultaneous excitation of DAPI, Alexa448, RRX and Alexa647 fluorophores. The system was

driven by Andor IQ3 software. Tiled imaging was performed to sample 2 mm$^2$ areas of skin. Stacks of 1 mm steps were collected with a 20x/0.75 CFI Plan-Apochromat air objective. Zen 2.3 software (blue edition, Carl Zeiss Microscopy GmbH, 2011) was used to stitch the acquired images. 40x oil objective was used to acquire z stacks of 0.5–1 mm steps.

## Developing hair-follicle density, and immunofluorescence quantitative analysis

Developing hair follicle density was measured from tiled images using Fiji software (NIH). Briefly: placode and cluster densities were quantified from 10 to 30 mm$^2$ regions of 14.5 back skins (*Apc*-null or *Apc*-het). For all the *Krt14rtTA* experiments, developing hair follicle densities were quantified across a total area of $\geq$8 mm$^2$ of E15.5 head skin, that is peak LV transduction. For each explant (n = 5 LGK1μM), we quantified the developing hair follicle density over a total area of 10 mm$^2$.

Placode and clusters morphological analysis was performed using the shape descriptors tool from Fiji (NIH). Area and circularity (*4pi(area/perimeter2)*) were measured and recorded. A circularity (cir) value of 1.0 indicates a perfect circle. As the value approaches 0.0, it indicates an increasingly elongated shape.

Intensity plots were generated like in *Messal et al. (2019)* using the plot profile tool from Fiji (NIH) and measuring intensities of a minimum of 3 basal cells per developing follicle (from a minimum of developing skin from three embryos). Briefly, optical sections of whole-mount 40x confocal images were converted into composite images in which MYC-tag (or FZD10) was in the red (or green) channel and DAPI (which labels the DNA) in the blue. Basal–apical intensities were measured along a straight line for each cell (and each channel) along the middle axis of the cell and normalized for intensity by subtracting the minimum value from each intensity profile and dividing by its average value. All measurements were aligned for the basal side of the cell having the same starting point of the measurement.

Transduction, ectopic and mispolarization expression efficiency were quantified using Fiji. Briefly, 40x optical sections of spinning disk confocal images were converte into composite images in which DAPI was in blue channel, H2BGFP in the green channel and MYC-tag in the red channel. From each optical section of a developing hair follicle, a minimum of 9 basal cells were quantified. The numbers of transduced cells (H2BGFP positive), ectopic expression (MYC-tag apical polarization) and mispolarization (basolateral expression of AQP4-MYC-tag) were recorded and the proportions calculated either relative to the total of basal cells analyzed (transduction) or the total number of MYC-tag expressing cells (ectopic and mispolarization experiment).

LEF1 immunofluorescence quantifications were performed using Fiji. Briefly, using spinning disk Z-stacks of whole-mount 40x confocal images, we sum the intensity across the follicle. The integrated density of LEF1 immuolabeling across a region of interest (e.g. dermal condensate or basal hair bud progenitors) was normalized at the cellular level by DAPI. Background was then measured and subtracted for each channel.

The relative population size of LEF1 or SOX9 was determined using Fiji. Briefly, using spinning disk Z-stacks of whole-mount 40x confocal images, we measured the area occupied by each specific cell population (LEF1 from dermal condensate, LEF1 from developing hair follicle and SOX9 hair follicle) and divided by the total epithelial area of the developing follicle, using the area tool.

## Statistics

To reduce any bias in data collection, all data from each group were not analyzed until all images were collected. No statistical method was used to predetermined sample size, randomization and experiment blinding was not used. Each experiment was repeated with at least two replicates and data presented are from three or more embryos, same age. Distributions were tested for normality using D'Agostino and Pearson test. To test significance, unpaired or paired two-tailed Student's *t*-tests were used for normal distribution and nonparametric Mann-Whitney test when the distribution did not follow a normal distribution. Basal to apical fluorescence intensity profile plots represent means and error bars SEM. Violin plots show the distribution of all measured data points. Median and quartiles are represented. All the other graphs represent means and error bars SD in all plots. Significance of *P* value was set at <0.05. Statistical details for each experiment, including the statistical test used, the sample size for each experiment, the n and *P* value can be found in the

corresponding figure legend. All graphs and statistics were produced using GraphPad Prism 8.2 for MAC, GraphPad Software, San Diego, California USA, www.graphpad.com.

## Acknowledgements

We thank JP Vincent for providing us with *Notum* floxed mice; Fuchs lab members M Nikolova, E Wong, L Polak, M Sribour for technical assistance and F G Quiroz, E Heller, S Ellis, A Sendoel, V Fiori, K Lay, R Adam, S Gur-Cohen for discussions; Rockefeller University's Flow Cytometry Core (S Semova, S Han, S Tadesse) for FACS sorting; Weil Cornell Medical College's Genomics Core for RNA-sequencing and raw data analyses.

## Additional information

### Competing interests

Elaine Fuchs: Reviewing editor, *eLife*. The other authors declare that no competing interests exist.

### Funding

| Funder | Grant reference number | Author |
| --- | --- | --- |
| National Institutes of Health | R37-AR27883 | Elaine Fuchs |

The funders had no role in study design, data collection and interpretation, or the decision to submit the work for publication.

### Author contributions

Irina Matos, Conceptualization, Data curation, Formal analysis, Supervision, Validation, Investigation, Visualization, Methodology, Writing - original draft, Project administration, Writing - review and editing; Amma Asare, Software, Formal analysis, Visualization, Methodology; John Levorse, June de la Cruz-Racelis, Methodology; Tamara Ouspenskaia, Investigation, Methodology; Laura-Nadine Schuhmacher, Resources; Elaine Fuchs, Conceptualization, Resources, Data curation, Supervision, Funding acquisition, Writing - original draft, Project administration, Writing - review and editing

### Author ORCIDs

Irina Matos  https://orcid.org/0000-0001-6100-8020
Tamara Ouspenskaia  https://orcid.org/0000-0002-5462-7103
Elaine Fuchs  https://orcid.org/0000-0002-0978-5137

### Ethics

Animal experimentation: All animal procedures used in this study are described in our #17020-H protocol named *Development and Differentiation in the Skin*, which had been previously reviewed and approved by the Rockefeller University Institutional Animal Care and Use Committee (IACUC).

### Decision letter and Author response

Decision letter https://doi.org/10.7554/eLife.54304.sa1
Author response https://doi.org/10.7554/eLife.54304.sa2

## Additional files

### Supplementary files

• Supplementary file 1. Key Resources Table.

• Transparent reporting form

## Data availability

RNA sequencing data have been deposited in the Gene Expression Omnibus under accession number GSE108745. All reagents engineered for this study are available from http://jdelacruz@rockefeller.edu under a materials transfer agreement with the Rockefeller University.

The following dataset was generated:

| Author(s) | Year | Dataset title | Dataset URL | Database and Identifier |
|---|---|---|---|---|
| Matos I, Asare A, Levorse J, Ouspenskaia T, dela-Cruz-Racelis J, Schuhmacher LN, Fuchs E | 2020 | WNT-signaling cells polarize inhibitors to protect their identity and fate | https://www.ncbi.nlm.nih.gov/geo/query/acc.cgi?acc=GSE108745 | NCBI Gene Expression Omnibus, GSE108745 |

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
