## [Decision Letter]

**Acceptance summary:**

This manuscript reveals that an internal gradient of WNT/WNT inhibitors drives 3D symmetry breaking in the hair placode. This innovative work will be of interest to developmental, stem cell, and skin biologists broadly.

**Decision letter after peer review:**

Thank you for submitting your article "Progenitors Oppositely Polarize WNT Activators and Inhibitors to Orchestrate Tissue Development" for consideration by *eLife*. Your article has been reviewed by two peer reviewers, one of whom is a member of our Board of Reviewing Editors, and the evaluation has been overseen by Marianne Bronner as the Senior Editor. The following individual involved in review of your submission has agreed to reveal their identity: Rui Yi (Reviewer #2).

The reviewers have discussed the reviews with one another and the Reviewing Editor has drafted this decision to help you prepare a revised submission.

Summary:

This manuscript analyzes how morphogens operate within closely-spaced, fate-diverging progenitors using transcriptomics and genetics. The authors show that Wnts and Wnt inhibitors to create boundaries and gradients to generate 3D pattern formation. Overall, this is an outstanding study that has a high impact for both WNT signaling studies and cell fate specification. The data are well controlled and very convincing. This work provides a plausible mechanism for directional Wnt activation in the hair placode.

Essential revisions:

1) Wnt activity in the placode: Both reviewers thought that the authors could provide a more robust measurement of Wnt activities to provide more insights into the proposed mechanism.

2) Protein expression: The analysis of Notum and Wif1 protein expression lacks quantification to show the localization of morphogens. Additionally, wider angle views of the Notum antibody data would document Notum and other gene expression patterns.

3) The deletion of APC, while impacts Wnt signaling, would also impact other aspects of cells including DNA damage and cell adhesion. Have that authors examined whether DNA damage and cell adhesion was altered in the APC mutant?

---

## [Author Response]

Essential revisions:1) Wnt activity in the placode: Both reviewers thought that the authors could provide a more robust measurement of Wnt activities to provide more insights into the proposed mechanism.

The reviewers felt we should add more evidence to validate our use of LEF1 as a proxy for WNT signaling. To this end, we now begin our Results section by citing our prior studies in which we have shown that in the skin, nuclear LEF1 colocalizes with a) TOPGAL, a WNT reporter driven by multimerized LEF1 DNA binding sites but requiring β-catenin for its action (DasGupta and Fuchs, 1999); b) nuclear β-catenin (Merrill et al., 2001); c) AxinLacZ a second WNT reporter, knocked into the Axin2 locus (Ouspenskaia et al., 2016). We also now add Figure 6—figure supplement 4 in which we add the lentiviral WNT reporter GFP, and show its co-localization not only with LEF1 but also with other products of WNT target genes like TCF1/TCF7, FZD10 and WIF1. We used LEF1 frequently because our goal in those experiments was to use its nuclear intensity to reproducibly measure how WNT signaling changed when we manipulated WNT inhibitors. Indeed we show that nuclear LEF1 in progenitor cells from the developing hair follicle robustly responded to the porcupine inhibitor treatment and wash-out. We now explain this more clearly in the text, and these new data and our discussion of prior data make this all the more forceful.

2) Protein expression: The analysis of Notum and Wif1 protein expression lacks quantification to show the localization of morphogens. Additionally, wider angle views of the Notum antibody data would document Notum and other gene expression patterns.

The reviewers also asked for data on endogenous NOTUM and WIF1 expression. We added a new figure (now Figure 4) where we show the pixel intensity profiles measured from immunofluorescence images using antibodies against the endogenous proteins. We validated the antibodies using KO tissue. Furthermore we show a wider view of the optical section. We use whole-mount imaging as it takes into account the whole structure and not just a snapshot of the tissue. The new figure illustrates the localization of WIF1 and NOTUM in placodes, buds and germs and underscores the reproducibility of the apical polarization of WNT inhibitors.

3) The deletion of APC, while impacts Wnt signaling, would also impact other aspects of cells including DNA damage and cell adhesion. Have that authors examined whether DNA damage and cell adhesion was altered in the APC mutant?

Finally, the reviewers asked about the impact of APC on other major cellular functions apart from WNT signaling. To this end, we’ve added 3 new supplementary figures and one table (Figure 1—figure supplement 2, Figure 1—figure supplement 4, Figure 1—figure supplement 5, Figure 2—figure supplement 2). To access for DNA damage we used immunostaining to detect the phosphorylation of the histone γH2AX that is present when double strand breaks occur. As judged by this assay, *Apc-null* cells were comparable to heterozygous and wild type (Figure 1—figure supplement 2). By contrast, both cell-cell and cell substratum changes were observed. E-Cadherin was lost altogether, but P-cadherin was retained (Figure 1—figure supplement 4). Given the pioneering studies of Takeichi showing that E-cadherin and P-cadherin cells sort out in a dish, these data explain why *Apc-*null cells cluster tightly. In addition, we observed a loss of the integrin β4, accounting for the loss of polarization of the *Apc-*null clusters (Figure 1—figure supplement 5). The new table shows upregulated transcripts that fall into the cell adhesion gene ontology category (Figure 2—figure supplement 2).